# Hitting the Target but Missing the Point: Recent Progress towards Adenovirus-Based Precision Virotherapies

**DOI:** 10.3390/cancers12113327

**Published:** 2020-11-11

**Authors:** Tabitha G. Cunliffe, Emily A. Bates, Alan L. Parker

**Affiliations:** Division of Cancer and Genetics, School of Medicine, Cardiff University, Cardiff CF14 4XN, UK; cunliffetg@cardiff.ac.uk (T.G.C.); batese@cardiff.ac.uk (E.A.B.)

**Keywords:** adenovirus, oncolytic, virotherapy, targeting, immunotherapy, immunogenic cell death, αvβ6 integrin

## Abstract

**Simple Summary:**

If harnessed appropriately, oncolytic viruses offer significant potential as anti-cancer agents. Such virotherapies can be engineered to replicate inside cancerous cells, stimulating the immune system, spreading daughter virions to surrounding cells and producing additional anticancer agents as a by-product of infection. To achieve this necessitates deep understanding of the biology of the virus and tumour cell, to tailor viruses from naturally pathogenic agents into refined, tumour selective “precision virotherapies” suitable for clinical translation. Here, we focus on the adenovirus, which in its pathogenic form causes transient and mild ocular, respiratory or gastrointestinal tract infections, depending on the serotype. We highlight advances that have been made in refining adenovirus to ablate natural means of infection and the strategies that have been employed to engineer viral tropism and selectivity for tumour cells. Further advances in these strategies will be required to deliver fully bespoke and efficacious precision virotherapies to the clinic.

**Abstract:**

More people are surviving longer with cancer. Whilst this can be partially attributed to advances in early detection of cancers, there is little doubt that the improvement in survival statistics is also due to the expansion in the spectrum of treatments available for efficacious treatment. Transformative amongst those are immunotherapies, which have proven effective agents for treating immunogenic forms of cancer, although immunologically “cold” tumour types remain refractive. Oncolytic viruses, such as those based on adenovirus, have great potential as anti-cancer agents and have seen a resurgence of interest in recent years. Amongst their many advantages is their ability to induce immunogenic cell death (ICD) of infected tumour cells, thus providing the alluring potential to synergise with immunotherapies by turning immunologically “cold” tumours “hot”. Additionally, enhanced immune mediated cell killing can be promoted through the local overexpression of immunological transgenes, encoded from within the engineered viral genome. To achieve this full potential requires the development of refined, tumour selective “precision virotherapies” that are extensively engineered to prevent off-target up take via native routes of infection and targeted to infect and replicate uniquely within malignantly transformed cells. Here, we review the latest advances towards this holy grail within the adenoviral field.

## 1. Introduction

Cancer treatment has come a long way in recent years, with 10-year survival rates increasing to around 50%, double that of 40 years ago [1]. While some of these improvements are credited to better and earlier diagnoses, a proportion of the advances in survival rates are attributable to the better understanding of cancer genetics, and thus how a patient may respond to a particular treatment. These advances have allowed clinicians to design and implement more efficacious and safer personalised treatment plans. Despite these advances, more progress remains to be made until fully personalised medicines are available for all patients. The need for such specific knowledge with a range of treatment options can lead onto the emerging era of targeted cancer medicines, in which the therapies act on a specific molecular target associated with the patient’s cancer [2]. Some targeted therapeutics are already being used as the first line treatment for patients in the clinic, such as the monoclonal antibodies Herceptin and the newer pertuzumab that target the receptor HER2, which is overexpressed in cancers such as metastatic breast cancer [3,4,5]. Herceptin treatment in patients with HER2 overexpressing tumours results in significantly better survival rates [6]. Despite these advances in targeted therapies, continued progress into the understanding of cancer-specific markers, such as upregulated HER2, and the development of targeted treatments are required to improve patient survival further.

### 1.1. Use of Targeted Therapies

Targeted therapies can come in many forms, from antibody treatments, such as Herceptin, or treatment involving inhibitors to important enzymes, including upregulated kinases, such as mitogen-activated protein kinase 3 (MAPK3), which has abnormal expression in many forms of cancer [2]. Small molecular weight cancer drugs are also important since they readily enter cells and affect changes compared to large molecular weight drugs such as antibodies. These inhibitors can be used to target and block many enzymatic pathways which support cancer progression. For example, vascular endothelial growth factor receptor (VEGFR) inhibitors which aim to reduce angiogenesis are currently being developed, with just under 20 different molecules being tested in in vitro and clinical trials [7].

### 1.2. Oncolytic Viruses

One rejuvenated area of research for targeted therapy is oncolytic viruses (OVs). These cancer-killing viruses exploit the natural ability of a virus to infect, replicate and lyse cells. An additional benefit of OVs is that the lytic nature of cell killing induces immunogenic cell death (ICD), increasing recruitment of immune cells to the tumour site. This ability of OVs to enhance host anti-tumour immune responses through ICD has the potential to turn immunologically “cold” tumours “hot”, thus sensitising otherwise refractory tumours to subsequent immunotherapies. This exciting potential of OVs to synergise with immunotherapies is the subject of significant ongoing clinical investigation, with compelling preliminary data obtained in several early phase clinical trials in brain tumours [8] and breast cancer [9], with other trials continuing in sarcoma, melanoma and breast cancer [10,11,12]. Efficacy may be further enhanced, with correspondingly reduced dose-limiting toxicities, through the development of rationally and effectively tumour targeted OVs suitable for intravenous applications.

The overarching aim in the development of OVs is to engineer viruses that can selectively target and/or replicate within cancer cells, leaving normal cells and tissues uninfected. The pro-immunogenic environment induced by OVs through the release of tumour antigens during cell lysis can be further enhanced by engineering the viral genome to over-express therapeutic transgenes, thus adding an extra layer to their therapeutic power. These multiple levels of activity are overviewed in Figure 1.

To limit “off target” uptake of OVs and subsequent killing of healthy cells, genetic engineering approaches have been undertaken. These have broadly focused on two approaches—firstly the active targeting of virions to tumour cells through the manipulation of viral capsid proteins to enhance uptake of virus into tumour cells and limit uptake via healthy cells. This has been largely achieved through the rational engineering of the viral genome and thus capsid to engineer tumour tropism via tumour associated antigens and receptors [13,14,15]. A second approach is to engineer in selectivity “post entry”, such that viral replication is blocked in non-transformed cells, but subtle modifications within the viral early genes permit replication to proceed within malignantly transformed cells. Using combinations of these approaches, it is possible to achieve tightly controlled tumour cell killing, and thus creating a more effective cancer treatment.

The first OV licensed in the western world by both the FDA and EMA is talimogene laherparepvec (T-Vec, Imlygic™), which exemplifies the early therapeutic potential of oncolytic viruses. T-Vec is a herpes simplex virus 1, with several genetic modifications. Deletions of the ICP34.5 gene reduces neurovirulence whilst also suppressing its ability to reactivate, which prevents it producing cold sores in the patient. A deletion in the ICP47 gene leads to enhanced antigen loading on MHC class I molecules. The MHC loading results in infected cells presenting tumour antigens, leading to enhanced immune recognition. These deletions improve the safety of this treatment as it limits the ability of the virus to completely evade immune responses. In addition, T-Vec is engineered to overexpress the cytokine GM-CSF (granulocyte-macrophage colony-stimulating factor) to further enhance immunogenicity through T-cell priming [16]. A truncated version of the US11 gene is also included to enhance the lytic activity of the virus by partial de-attenuation [17]. The virus takes advantage of disrupted anti-viral pathways in cancer cells to enable selective virus replication. One such pathway is the PKR (protein kinase R) pathway which is key in regulating cell proliferation. Normally, this pathway is activated by the dsRNA produced when the virus replicates, thus triggering protein synthesis inhibition. However, in cancer cells with aberrant regulation, the dsRNA production warning sign is ignored, allowing the virus to replicate, leading to viral propagation and cell lysis. Disruption in the PKR pathway has been found in approximately 70% of melanoma cells, and thus T-Vec is a good replication selective therapy for melanoma cancers [18,19].

T-Vec was licensed for localised treatment of recurrent melanoma [20,21]. In phase III trials using T-Vec to treat unresectable late stage melanoma, durable response rates of 16% vs. 2% were shown, compared to control group, increasing the survival rates of these patients [22,23]. Moreover, recent studies have demonstrated that virotherapies such as T-Vec can be a powerful tool in adjuvant therapy and have been used to sensitise triple negative breast cancers to follow up treatment with immunotherapies [9,24].

### 1.3. Adenovirus as an Oncolytic Virotherapy

Despite the success of T-Vec, the therapy is only licensed for local intra-tumoural applications. Therefore, its use for treatment of metastases, which kills the majority of cancer patients, is limited. Metastasis treatment would require blood system disseminated therapy to individually target each lesion through intravenous (IV) delivery, or utilising the abscopal effect from treatment of one or more lesions, which could lead to a systemic anti-tumour immune response extending to the milieu of micro metastatic deposits in the body [25,26,27]. Therefore, significant research is ongoing into other viral vectors that may be better suited to IV delivery, including adenoviruses (Ads) [28], reovirus [29] and vaccinia virus [30]. Ads have been shown to be the most durable of these options and are the most studied clinically and experimentally. Ads also have the advantage of being naturally lytic, immunogenic and can be produced to high titres and purity, all important considerations when developing such therapeutics for widespread clinical application.

#### 1.3.1. Adenovirus Cell Entry and Trafficking

Human adenoviruses can be classified into seven species (termed A–G) comprised of over 100 serotypes [31]. Ads commonly infect the respiratory system however different serotypes can also infect the gastrointestinal, cardiac, neurological, ophthalmological and genitourinary tissues which can result in an array of clinical pathologies [32]. Ads bind to receptors on the membrane surface. Ad5 has been well described as recognising the coxsackievirus and adenovirus receptor (hCAR) [33]. Other receptors involved in adenoviral attachment are CD46 [34], desmoglein 2 (DSG2) [35] and sialic acid [36]. Primary receptor binding is dependent on the Ad serotype but generally species A, C, E and F interact with hCAR while species B and D utilise other receptors [37]. Species B is reported as using primarily CD46 and DSG2, and, although the individual receptors for many species D serotypes are unknown, it is thought there is sialic acid involvement [38,39]. Initial receptor attachment is followed by internalisation mediated by αvβ3/5 integrin binding through a conserved Arg-Gly-Asp (RGD) motif [40]. Upon cell entry, the adenovirus is partially disassembled via endosome acidification and the uncoated is released and transports viral DNA into the nucleus [41]. Viral replication takes place in two phases: early phase and late phase. An overview of the adenoviral replication cycle is illustrated in Figure 2.

#### 1.3.2. Oncolytic Adenovirus

Many Ad-based therapies in the clinic currently use replication-based control as a mechanism for cancer targeting. They are reliant on cancer selectivity at the point of cellular replication at a post cell entry stage rather than bona-fide tumour tropism. The concept that adenovirus infection mirrors several key hallmarks of cancer [45] underpins the mechanism that restricts the replication of many early oncolytic adenoviruses to tumour cells [46]. An oncolytic adenovirus that preferentially replicates in a tumour cell environment often takes advantage of genes that are frequently overexpressed in cancer. Deletion of viral replication genes to render the virus replication-incompetent in healthy cells, but replication proficient in tumour cells with dysregulated or inactivated tumour suppressor gene function [47,48]. These are termed replication selective oncolytic or conditionally replicating adenovirus [49,50].

Adenovirus have many benefits for use as oncolytics. These include their relative ease of manipulation, being double stranded DNA viruses. Their capacity for transgene incorporation, being around 6 kb for non-replicating vectors and around 2 kb for oncolytic vectors is more than sufficient to encode therapeutic transgenes (or combinations of transgenes) to enhance the pro-immunogenic tumour microenvironment. These transgenes could include immune checkpoint inhibitors targeting PD-L1 or CTLA4 [51,52], cytokines such as TNFα and IL-2 [53] or chemokines including CCL5 [54]. Alternatively, genes encoding proteins that are directly cytotoxic to the tumour cells such as REIC/DKK-3 can also be incorporated [55].

Although early clinical data for many oncolytic viruses are encouraging, the exact mechanism of cell killing often remains unclear [56]. It is evident the viral and host cell interactions are complex, particularly in the context of systemic delivery and within the tumour microenvironment, and an understanding of the tumour and virus biology will provide insight and enhance future oncolytic virotherapies [57]. The popularity of adenovirus is evidenced by the sheer number of clinical trials, standing at 237 at the time of writing, that use adenovirus for cancer treatment in some form [58]. These trials have demonstrated safety and feasibility; however, delivery and efficacy must be improved if oncolytic adenovirus is to achieve its full promise as an effective cancer therapy [59,60].

Despite their immense potential, adenoviruses, especially those based on the species C serotype Ad5, have several pitfalls which need to be carefully addressed to tailor the OV in to an effective therapeutic. These disadvantages include the high rates of pre-existing immunity against Ad5 in the populations where Ad5 is a common pathogen. These levels of pre-existing immunity vary geographically from >90% in sub-Saharan Africa [61] to ~30% in the UK population [62]. High levels of pre-existing immunity will promote the rapid removal and destruction of the therapeutic by the reticuloendothelial system, resulting in limited bioavailability for active tumour targeting [63]. A further limitation stems from the native infectious routes via the capsid proteins of Ad5 that can also cause dose-limiting interactions and toxicity (Figure 3). The widespread anatomical expression of the primary receptor, Coxsackie and adenovirus receptor (hCAR) [33], means vectors based on Ad5 will be sequestered and infect a wide range of off target (non-cancerous) tissues in the body [64] or may become irreversibly trapped in the blood [65]. One way this can be overcome is by genetic modification of the amino acids 408 and 409 within the AB loop of the fibre knob protein (Fkn) to remove binding to hCAR (called the KO1 mutation) [66]. Although Ad5 predominantly uses hCAR there are alternative receptors utilised by other species including CD46 and desmoglein-2 (DSG-2) which are the primary entry route for Species B adenovirus [34,35]. The species B Ad3 pseudotype is a prominent oncolytic virus which uses both CD46 and DSG-2 for cell entry. CD46 is expressed on almost all nucleated cells, and DSG-2 is a cardiomyocyte [67] and tight junctions restricted receptor [68], and therefore present additional considerations for ablation of native binding tropisms. Species D does not appear to bind these three known adenoviral receptors with any significant affinity. There is evidence to suggest these viruses may be more likely to use sialic acid as their mode of entry [36,37,69].

Other capsid proteins can also cause off-target binding and sequestration issues. The hexon protein of Ad5 binds with high affinity to the blood clotting factor X (FX), which results in rapid and efficient transduction of hepatocytes, with consequent potential hepatotoxicity resulting from of Ad5 vectors [70,71]. The penton base protein on the capsid also has implications for off-target effects. The RGD domain in the pentameric protein group binds to integrins αvβ3/αvβ5 leading to downstream signalling for internalisation [72]. These interactions are also thought to lead to uptake in the spleen inducing consequent pro-inflammatory responses against the Ad [73,74]. Therefore, mutation within the RGD binding region in the penton may be important in limiting these off-target effects [75].

In this review, we discuss the current approaches and significant refinements to the Ad5 capsid necessary to prevent off target interactions. We also consider alternative approaches to circumvent the Ad5 associated limitations and generate precisely guided cancer therapeutics.

## 2. Genetic Engineering of Oncolytic Adenovirus

The adenoviral genome is organised into early (E) and late (L) genes (Figure 4). The early phase genes encode proteins that regulate the host and viral proteins, avoid premature cell lysis and prepare components for DNA replication. Late phase produces structural proteins that are required for the assembly of mature virions [44].

The standard approach in the design of novel oncolytic virotherapies involves making modifications viral genes to improve cancer cell selectivity and oncolytic potency. The ability to engineer the double stranded DNA genome of adenovirus with relative ease has been proven for clinical applications from vectors for gene therapy and vaccines to oncolytic viruses [76]. A key feature in development of adenoviral vectors are the modifications to reduce the immunogenicity and bypass innate anti-viral immune responses.

First generation adenoviral vectors harbour deletions in the E1 and E3 regions [77,78]. These deletions not only improved the vector safety profile but also create significant space necessary for the insertion of transgenes [79]. The E1 genes encode proteins necessary for viral replication, therefore E1 deletion results in a replication deficient virus [44]. Consequently, vectors with this deletion must be propagated in cell lines expressing E1 products in trans, such as 293 or PER.C6 cells [80,81,82]. E3 encoded viral proteins are involved in evading host antiviral immunity but are not essential for viral replication, deletion of this region allows insertion of larger genes but may reduce the oncolytic potency [83]. Second generation adenoviral vectors may also have the E2 and E4 regions deleted which eliminates expression of most Ad genes and allows more room for transgene insertion [84]. However, these vectors must be propagated in cell lines that express E1, E2 and E4 gene products. The late genes are involved in structure and therefore are required for production of mature virions. A final generation of Ad vectors that are lacking all viral coding regions have been developed. These are termed gutless or helper-dependent Ads as they require co-infection of a wild-type adenovirus or helper vectors [85]. These have promising therapeutic advantages but are difficult to manufacture in high quantities.

Conditionally replicating adenoviruses (CRAds) encompass several oncolytic adenovirus therapies in the clinic. They can be classified into two types, however both approaches involve modifications in the E1 region of the adenoviral genome. Adenovirus E1 is comprised of two genes: early region 1A (E1A) and early region 1B (E1B). E1A is the first transcribed gene post infection and promotes progression into S-phase of the cell cycle. E1B encodes genes that protect the cell from undergoing apoptosis as a result of E1A induction of S phase and enables the virus to undergo productive replication in the host cell. The first strategy employed when designing CRAds is to replace the E1 promotor with a tumour specific promotor, therefore preventing induction of E1A mediated viral replication in the absence of the appropriate promoter [86,87,88]. This approach can be used to restrict replication and to start the expression of the treatment transgenes within tumour cells. One example is the promotor survivin, which has been used to this effect, regulating the expression of the heat shock protein 70 (Hsp70) that inhibited tumour growth in gastric cancer and adult T-cell leukaemia (ATL) [86,89]. Another promotor of note is human telomerase reverse transcriptase promotor (hTERT). This promotor can enhance cell lysis, leads to increased release of viral progeny for further infection and shows reduced hepatocyte effects compared to ONYX-015 in solid tumour in-vivo models [90].

The second strategy relies on modifications within the E1 region preventing the virus from restricting host cell defences (for example, pRb mediated apoptosis) and therefore the virus is only able to replicate in tumour cells defective in these pathways. One of the most effective mutations described to date is the dl24 (∆24) mutation. This mutation is a 24-base pair deletion in the constant region of E1A gene. This deletion is in the region that is responsible for binding the Rb protein and so targets replication to cells with abnormal Rb control that can bypass this pathway. This leads to selective replication in cells that are defective in the Rb/p16 pathway, which has been identified in the majority of cancers, including gliomas and ovarian cancers [91,92,93,94].

Another mutation used is the T1 mutation, which has a truncating insertion in the E3/19K protein. The T1 mutation means that this protein is relocated to the plasma membrane and enhances the release of virus from infected cells [95]. Therefore, this mutation may be a useful addition to a tumour-selective Ad-based therapy, such as in the oncolytic ORCA-010 [96].

## 3. Current Clinical Applications of Oncolytic Adenoviral Therapies

ONYX-015 (also referred to as ∆1520) was one of the first replication selective oncolytic adenoviruses tested in a clinical setting for the treatment of head and neck cancer [97,98]. ONYX-015 harbours an E1B55K deletion, which was originally thought to be essential for viral replication as it sequesters p53 and promotes cell cycle transition [99]. p53 is commonly lost or downregulated through mutations in multiple cancer cells [100], and it is therefore considered that ONYX-015 would replicate almost exclusively in cancer cells lacking p53. However, more recent research suggests that this mutation is more likely to work through the loss of late viral RNA export, rather than through p53 status alone. The mechanism of action may be more complex than originally thought, likely due to multi-modal action of the p53 pathway [49,101,102]. Whilst ONYX-015 and a variant H101 (E1B55K and E3 deletion) demonstrated the safety of oncolytic adenoviruses, the efficacy was limited by attenuated viral replication and spread [96,103]. Subsequent generations of tumour selective oncolytic adenoviruses contain mutations in the E1A gene that functions through binding the retinoblastoma protein (pRb) [91]. Several oncolytic adeno-virotherapies have since entered clinical trials (Table 1) and have demonstrated safety and feasibility. However, delivery and efficacy must be improved if oncolytic adenovirus is to be used as an effective cancer therapy, especially as a systemically administered agent capable of effectively targeting tumours and metastases [59,60].

The vast majority of adenovirus research to date has focused on the species C, Ad5. Several oncolytic adenoviruses (OAds) have demonstrated limited efficacy in clinical trials as a result of poor viral persistence [104,105]. Although this is, in part, related to the early design of these viruses, it may also result from high levels of pre-existing immunity [106]. A substantial proportion of the population will have experienced an acute adenovirus infection, and many will have developed neutralising antibodies against the most common Ad serotypes [32]. Activation of anti-tumour immunity whilst dampening the innate host anti-viral immune response is essential to the success of OAds. An alternative approach, therefore, may be through the development of alternative, low seroprevalence adenoviral species, such as those from Species B or D, which tend to have naturally low levels of pre-existing immunity in the population [107]. Such serotypes may also exhibit naturally lower levels of off target uptake due to reduced interactions with components of the blood. Ad5 is known to bind to FX in serum which mediates sequestration by the liver and can impact virus delivery to the tumour [108]. FX binds the hexon protein on Ad5 capsid and can result in off target uptake [109]. It was observed that alternative species, for example species D Ad26, does not bind FX in the same manner as Ad5 [109]. This knowledge was used to identify key residues in FX binding through sequence alignment and has fed into the production of retargeted Ad vectors however the use of alternative species as an alternative to Ad5 has not been fully explored [71]. The use of novel oncolytics developed from rarely isolated serotypes from species D may represent an exciting and alluring possibility, where the diverse nature of this species represents a significant and largely untapped repository for investigation. Recent significant progress has been made in this regard by the Ehrhardt laboratory, who have investigated a larger spectrum of adenoviral vectors and begun to evaluate their potential for oncology applications [110,111].

Alternative adenoviral species offer many advantages over the commonly used Ad5-based therapies however, despite their increasing popularity as platforms for vaccine applications, they are poorly represented in the clinical oncology setting. Ad5/kn3 pseudotype has demonstrated some limited efficacy in ovarian cancer [112,113]. This targets via species B receptors which are not cancer specific and maintain the previously outlined issues associated with Ad5-based vectors.

Species B adenovirus have demonstrated the potential to play an important role in the field of adenoviral oncolytics. An alternative strategy to the rational development of novel Ad serotypes for oncology applications has been to develop a novel chimeric OV through a process of natural selection for recombinants with enhanced cell killing activity in cancer cells such as with enadenotucirev (EnAd, formally known as ColoAd1; PsiOxus Therapeutics Ltd., Abingdon, UK). Several adenovirus serotype recombinants were selected for on colon cancer cells (HT29) by this method of “directed evolution” (Figure 5) [114]. Despite the initial pool representing diverse serotypes from species B–F, the resultant chimera is a fully species B recombinant, derived from Ad11p and Ad3 [114]. This virus demonstrated potency and selectivity greater than ONYX-015 and Ad5 [115]. The tumour selectivity mechanism for EnAd is not fully understood however initial clinical trials have demonstrated durability and tolerability, not only for colorectal cancers but with other solid cancers [115,116]. EnAd is thought to act through a non-apoptotic mechanism termed ischemic cell death and possesses pro-inflammatory properties [117]. Species B therefore represents an exciting avenue for development of oncolytics, however the lack of intrinsic tumour specificity and high prevalence of species B receptors, CD46 and DSG-2, on healthy cells is an important consideration that may result in the depletion of virotherapy available for active targeting of tumours through off-target sequestration. Ablation of these native interactions or use of Ads that bind receptors with weak affinity may result in improved novel retargeted oncolytics.

## 4. Detargeting Ads

Adenovirus has not evolved as a cancer selective pathogen, and therefore requires engineering in order to effectively target cancer cells. In Ad-based therapies, the efficiency of the treatment can be greatly affected by their native binding interactions. The most studied Ad, Ad5, binds hCAR which localises to the tight junctions between cells and is expressed ubiquitously throughout the body [33,118]. hCAR has also been reported as downregulated in certain cancers [119,120,121]. Therefore, reliance upon hCAR as an entry receptor for any Ad-based cancer therapy would greatly limit its uses, as transduction would be limited to cancer cells with high-hCAR expression, and off-target transduction could lead to tissue toxicity. Moreover, more aggressive cancer growth correlates with loss of hCAR expression, and so non-targeted Ad therapy is unlikely to treat aggressive cancers through hCAR transduction alone [118].

hCAR binding can be ablated by mutating the key amino acids in the fibre knob AB loop (L5 gene), using the KO1 mutations, S408E and P409A [66,122]. hCAR mediated cell entry is a two-step mechanism. First the virus attaches to hCAR and, secondly the virion internalised through binding of an Arg-Gly-Asp (RGD) motif in the penton base to αvβ3/αvβ5 transmembrane integrins on the cell surface [123]. This secondary interaction has been exploited to further detarget Ad5 through the RGD to RGE modification of the penton base and improve cancer targeting. The Ad5_NULL_ vector encompasses the KO1 mutation, the RGD to RGE modification as well as a modification within the hexon hypervariable region (HVR7) in order to ablate binding to coagulation FX and prevent sequestration by and transduction of liver hepatocytes [13,124].

Larger modifications can also be made to detarget the virus particle, such as replacing the fibre shaft of Ad5 with shorter variations found in other serotypes such as Ad40, or Ad41. This approach has shown reduced binding to cells [125,126,127].

Chemical modifications can also be used as a means of detargeting through polymer coating of the Ad particle. Polyethylene glycol (PEG) is commonly used for this purpose, due to its cationic properties [128]. The main advantage of coating the Ad is to prevent neutralisation by pre-existing antibodies which reduces the efficiency in patients previously exposed to the Ad. The use of chemical modifications was well reviewed elsewhere previously by Kreppel and Kochanek [129] and Kim et al. [130]. A major disadvantage of non-genetic means of targeting is that daughter virions produced through replication will not harbour the modifications necessary to target tumour cells, and therefore genetic strategies which are heritable are therefore more commonly preferred.

## 5. Retargeting Strategies

Here, a range of retargeting strategies is discussed, and both CRAds and oncolytic virus results are considered in this section together. This is due to the abilities of the targeting strategies to be applied in either context.

### 5.1. Pseudotyping

One relatively common method to introduce new tropism to an Ad-based therapeutic is to use chimeric fibre knob/shaft proteins through pseudotyping. Pseudotypes are recombinant adenoviruses that combine different aspects and structural proteins from differing serotypes into a chimera, and are often generated to cherry pick optimal features associated with different serotypes [131]. This genetic strategy uses Fkn (fibre knob) proteins from less-commonly used Ad species that do not use hCAR as a primary transduction mechanism and substituting these onto an OV based on Ad5. This confers new binding abilities without having to move away from the large knowledge base of commonly used Ad5. This has shown some success in colon cancer [132] and ovarian cancer [133] amongst others.

The use of chimeric Fkns (cFkns) has been extended by sequences from non-human Ad species [134] and through chimeric fibres created from several sequences such as bacteriophage T4 fibritin and human CD40 ligand (CD40L) in conjunction with the Ad5 Fkn [15].

Pseudotyping can also be used for the whole fibre protein comprising both the shaft and knob domain into the Ad5 capsid. However, the “tail domain” is an important consideration in this strategy, as the maintenance of a portion of the N-terminus of the parent fibre shaft is required for the translation of binding to the penton base of the parental capsid [135]. Shayakhmetov and Lieber demonstrated that pseudotyping the fibre proteins can also alter the binding capabilities of the virus. Pseudotyping the fibre shaft and the Fkn proteins from Ad5, Ad9 and Ad35 results in different receptor usage and intracellular trafficking, partly down to the length and geometry of the fibre shaft [136].

Though this method has shown promising results, it has its limitations. Creating a chimera that can correctly fold once it is attached to the fibre shaft can be a limitation in itself. The cFkn formed and its binding capabilities are likely dependent on its ability to form trimers, which is required for native Ad binding [137]. If such a protein is found that can correctly fold once attached to the fibre shaft, and form the trimers required for binding, it then must confer a novel binding tropism. The natural array of binding tropisms already understood from the less-commonly studied Ad species that can be pseudotyped is limited. Therefore, although this method has its place as a tool for investigating the tropism of rarely isolated adenoviral species, other mechanisms may prove more useful in the context of developing tumour tropism.

### 5.2. Peptide Retargeting

One mechanism employed experimentally to enhance hCAR independent uptake of Ads into cells is to enhance targeting to upregulated αvβ3/5 integrins on tumour cells. The most successfully deployed has been the RGD-4C motif, incorporated into the HI loop of the fibre knob, which has demonstrated improved uptake in cancers expressing high levels of integrins such as ovarian cancer and glioma [138,139]. This modification enables suitable presentation of the integrin interacting RGD motif, held in position by the pair of disulphide bonds between the cysteine residues, to successfully engage with cellular integrins and stimulate uptake via endosomes.

Other retargeting methods have also proven effective, such as insertion of peptide sequences which confer a known binding ability within the Fkn protein, though this comes with its own limitations. The sites within the Ad5 fibre knob domain in which peptide sequences can be inserted successfully have been narrowed down through structural studies, demonstrating the HI loop and the C-terminus of the protein as the most promising sites [123,140]. Within other serotypes, hypervariable nature of the loops within the fibre knob protein have demonstrated that other loops are more surface exposed and thus better suited to genetic insertions, for example the DG loop in Ad48 has been shown to be the region best suited to genetic manipulation [141,142]. Insertions in these sites have shown promising results in targeting Ads, such as targeting towards ovarian, breast and prostate cancer cell lines by insertion of Her2/neu-reactive Affibody into the HI loop of a native-binding ablated Ad5 vector [143,144].

Peptide sequences from other viruses had also been shown to target an Ad-based therapy towards cancer cells. Insertion of a 20-aa peptide, NAVPNLRGDLQVLAQKART, native to foot and mouth disease virus (FMDV) was identified as a binding peptide sequence to αvβ6 integrin [145], an integrin that is reported to be upregulated in certain epithelial cancers, including breast, ovarian, pancreatic and colorectal [146,147,148]. Figure 6 illustrates the Ad5 fibre knob protein engineered to present the A20 peptide within the HI loop in complex with αvβ6 integrin. This peptide has previously been studied in terms of cancer research, used for non-invasive radiolabelled peptide for cancer imagery, whilst antibodies and CAR-T cells directed to αvβ6 integrin are also being investigated, underpinning the potential of this biomarker for cancer selective targeting [145,149,150]. The A20 peptide has been successfully incorporated into the Ad5_NULL_ to create the Ad5_NULL_-A20 OV, which is a highly selective virotherapy targeting ovarian cancer [13], and with significant promise to target other epithelial cancers expressing high levels of αvβ6 integrin.

Other sites in the capsid that are amenable to insertions have also been explored, such as the hexon gene. Insertion into the hexon has good potential for targeting due to its abundance in the capsid. If all copies displayed the insertion, it could lead to a coated Ad capsid with 720 copies of the targeting peptide. This has shown some recent success, through insertion of muscle binding peptides into the hypervariable region 4 (HVR4) in the hexon protein [14], as well as the RGD-4C peptide [151,152].

When modifying viral capsid proteins, additional considerations should be made. The size, structure and charge of the insert can affect its success due to steric hindrances. The rate-limiting step is the lack of efficacious tumour targeting peptides that can be incorporated into the viral capsid efficiently to retarget towards cancer cells with limited off-site effects.

Targeting peptides which have proven effective when presented within the context of the three-dimensional fibre knob protein often appear to be those which have a degree of secondary structure. The secondary structure is important to consider as strategies using linear peptides have proven to be less successful. For example, the RGD-4C peptide is designed to present the RGD integrin interacting tripeptide at the apex of the loop, held in place by disulphide linkages. Similarly, the A20 peptide forms an alpha helical confirmation both in its native context within the vp1 protein of FMDV, which it retains when transferred into the Ad5 fibre knob protein, thus retaining the geometry required for receptor engagement. For future targeting strategies to be successful, it will likely require the development of sophisticated molecular technologies, capable of high throughput evolution, screening and selection of knob variants with increased binding affinity for tumour associated antigens of interest.

### 5.3. Techniques for Targeting Peptide Discovery

Other methods for retargeting Ads have also had some success. Despite the success of Ad5_NULL_-A20, there has been limited continued success using the Ad5_NULL_ platform to target other tumour-associated antigens (TAAs) and receptors. The rate-limiting step is the lack of efficacious tumour targeting peptides that can be incorporated into the viral capsid efficiently to retarget the Ad5_NULL_ platform towards tumour cells. Previously, we and others have utilised methods such as bacteriophage (phage) biopanning to identify peptides that can bind TAAs [153,154,155].

Biopanning is an approach that uses affinity-based selection. Random peptide libraries can be created and displayed on the phage, often in the pVII or pIII gene of filamentous phage M13 (Figure 7). M13 has around 5 copies of each pVII/pIII gene products in the capsid, located at the end of the cylindrical phage. The resultant library is incubated with a target protein (or cell line), allowing binding to occur. Unbound phage, or those with low affinity, are then washed away. Finally, those random peptides still bound strongly to the target are eluted, either by changing the pH or by competitive inhibition. The process is repeated to identify peptides with the highest affinity for the given target, which can be sequenced for further use [156]. Phage display allows for high throughput analysis of peptide libraries for targeting a specific receptor protein. In fact, this method has been used frequently, with peptides targeting EGFR [157] and HER2 [158]. Promising peptides can be inserted into the permissive regions of Ad5 Fkn [155]. This has been tried with several different peptides targeting cancer-specific markers, such as folate receptor α (FRα) commonly upregulated in ovarian cancers [159]. However, after binding, the FRα mediated cell entry mechanism does not allow for correct intracellular trafficking, showing retention of targeted virotherapies in late endosomes in FRα positive ovarian cancer cells, with limited successful transduction [153]. This highlights another potential consideration when developing targeting approaches for adenoviral-based oncolytics—not all TAA receptor pathways will be compatible with clathrin-mediated endocytosis pathways and thus represent viable entry routes for adenoviral-based virotherapies.

Additional limitations to this approach revolve around the linear orientation of the peptides being selected and displayed. Although promising peptides can be found through this approach, once they are incorporated back in the Fkn, the peptide can change confirmation due to the three-dimensional nature of the viral capsid protein into which the peptide is engineered. It could be assumed that the difference, and therefore the success, of the A20-peptide is due to the constrained orientation was maintained. A20 was identified in FMDV and transposed into Ad, and thus the orientation was conferred. Conversely, in phage-display technology, the selection is based around incorporation into the capsid coat proteins in a linear orientation.

A secondary issue of note for the future of this technology is that insertions can only be made in one linear string of DNA, creating one addition to the coat proteins in the phage particle to be used in selection. For this technology to be the answer to extending the use of OAds to many different cancer types, with very different surface protein expression profiles, there needs to be a way of incorporating, and thus selecting for, multiple regions that confer binding but are not next to each other in linear DNA sequence. This would create additional problems such as the requirement for multiple incorporation sites in the Ad Fkn protein and the complexity of maintaining the correct confirmation of these multiple sites in protein space (for example, distance apart on the Fkn, interactions with polar amino acids nearby limiting availability for binding and flexibility of the insert). However, if these were to be achieved, it would allow for quick, efficient and effective selection of cancer-specific binding peptide regions for incorporation into an Ad-based vector.

Lupold et al. also developed a useful technique for retargeting, using the Ad particle itself. The pTex system uses a similar approach to pseudotyping mixed with peptide insertions, but allowing isolated and randomised mutation of the fibre knob for later incorporation into the capsid. This system overcomes some of the issues, such as linear display, although it may be limited for targeting towards a specific receptors of interest without the issue of previously understood peptide-ligand interactions [160].

The detargeting and retargeting methods highlighted in this review are overviewed in Figure 8.

## 6. Conclusions

Oncolytic adenoviruses are powerful therapeutic agents with great potential in the clinical arena, combining multiple cell-killing effects on the tumour microenvironment. Firstly, the life cycle of adenovirus induces immunogenic cell death. Moreover, the process of replication and lysis results in the production of many tens of thousands of additional daughter virions, which when released, infect surrounding cells, thus repeating and amplifying the process. Additional engineering of the viral genome to encode therapeutic transgenes, such as immunotherapies, cytokines or pro-apoptotic proteins can further enhance the immunogenicity of the tumour microenvironment effectively turning the tumour into a factory producing protein to promote its own destruction.

Despite some early evidence of efficacy as a combination therapy in the clinic, the development of fully refined oncolytic adenoviruses has failed to reach its full potential. There are numerous obstacles to sequentially consider when developing novel adenovirus based oncolytic virotherapies, including infection of non-cancerous cells, activation of the anti-viral immune response and a limit in the number of viruses with native cancer tropism. The ability to modify the adenoviral genome and overcome these limitations makes them attractive candidates for targeted oncolytic virotherapies. Furthermore, there is a vast repository of alternative adenoviral serotypes, possessing known advantages over Ad5-based therapies, that are yet to be explored in an oncology setting. Employing techniques such as peptide insertion has had promising pre-clinical results and if combined with additional modifications to further detarget and arm with therapeutic transgenes, the result would be highly potent targeted oncolytic virotherapies. Whilst significant progress has been made in developing such systems (e.g., the Ad5_NULL_ platform), step changing technologies will be required to develop optimally targeted “precision virotherapies” to tumour specific molecular addresses, and thus to deliver truly personalised virotherapies moving forwards.

Therefore, the remaining limitations for targeted oncolytic applications using this approach are the identification of ligands that are cancer-specific [161], coupled with the poor ability to transfer linearly selected peptides from phage libraries into the three-dimensional Ad structure. In a sense, whilst technologies exist to elucidate peptides or antibody fragments that allow us to “hit” tumour targets of interest, the success of such targeting technologies when translated into oncolytic virotherapies will require smarter systems, designed to engineer tropisms directly into the viral capsid protein of relevance to be successful, or else they will continue to “miss the point”. Developing technologies deigned to overcome these limitations will be key to the future success and efficacy in the clinic.

## Figures and Tables

**Figure 1 cancers-12-03327-f001:**
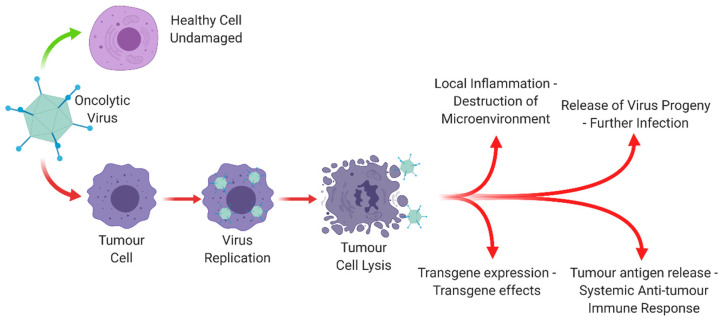
The mode of action of an oncolytic virus. Oncolytic viruses leave healthy cells undamaged, whilst leading to a range of effects in tumour cells which lead to lysis, further infection and an immunological response. Created with https://biorender.com.

**Figure 2 cancers-12-03327-f002:**
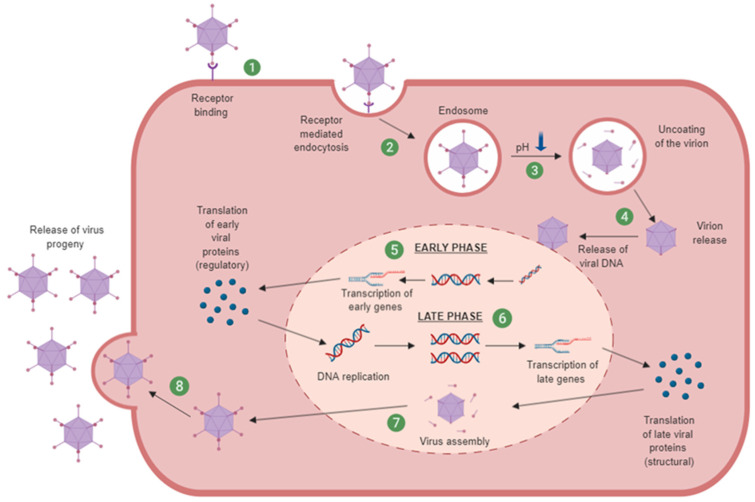
Adenovirus replication cycle. (**1**) Virus attachment to receptors on the host cell surface. (**2**) Internalisation of the virus by endocytosis. (**3**) Low pH results in endosomal acidification and partial disassembly of the virion. (**4**) Virion released from endosome and trafficked to the nuclear pore complex where it releases viral DNA into the nucleus. (**5**) Early phase: Transcription and subsequent translation of early genes to the regulatory early proteins. (**6**) Late phase: Transcription and subsequent translation of late genes to the late structural proteins. (**7**) Assembly of progeny virion. (**8**) Cell lysis resulting in release of mature virus. Created with https://biorender.com. Figure adapted from [41,42,43,44].

**Figure 3 cancers-12-03327-f003:**
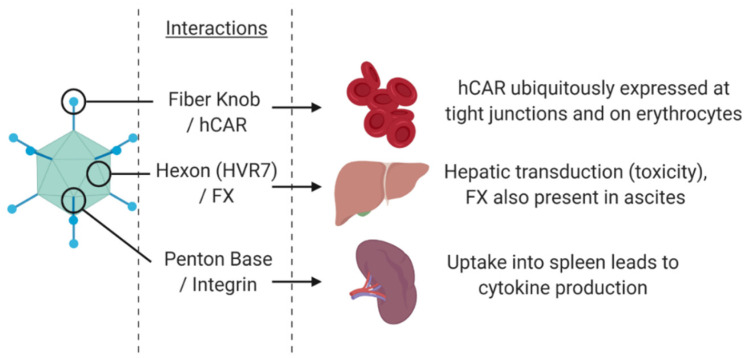
Dose-limiting Ad5 interactions in vivo. The fibre knob protein binds to hCAR expressed at tight junctions and on erythrocytes, the hexon binds to Factor X (FX) in the blood and the penton base binds to αvβ3/5 integrins. These binding interactions would lead to off-target effects. Created with https://biorender.com.

**Figure 4 cancers-12-03327-f004:**
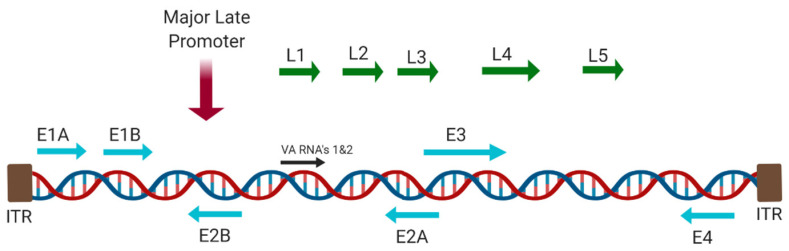
Adenoviral genome, highlighting key genes which are often modified or deleted in oncolytic therapeutics. Created with https://biorender.com.

**Figure 5 cancers-12-03327-f005:**
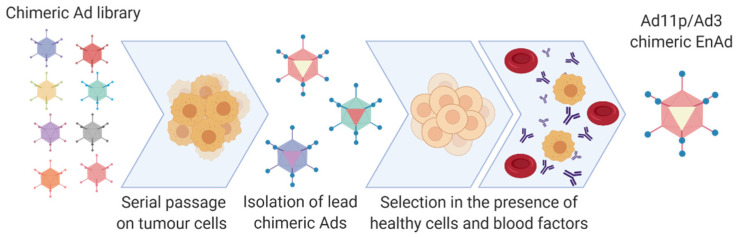
Method of production for oncolytic virus, EnAd (PsiOxus Therapeutics Ltd., Abingdon, UK). Serial passages of an Adenoviral library in tumour cells lead to chimeric Ads development. Selection with healthy cells and blood factors removes those that bind off-target receptors, lead to Ad11p/Ad3 chimeric EnAd being selected for. Created with https://biorender.com.

**Figure 6 cancers-12-03327-f006:**
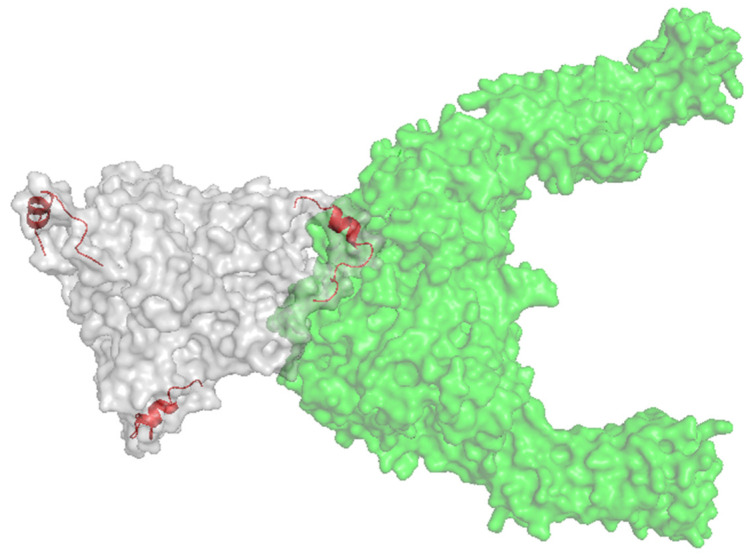
Ad5 knob (white) with the A20 peptide insertion (red) in complex with αvβ6 (green). Image created using PyMol.

**Figure 7 cancers-12-03327-f007:**
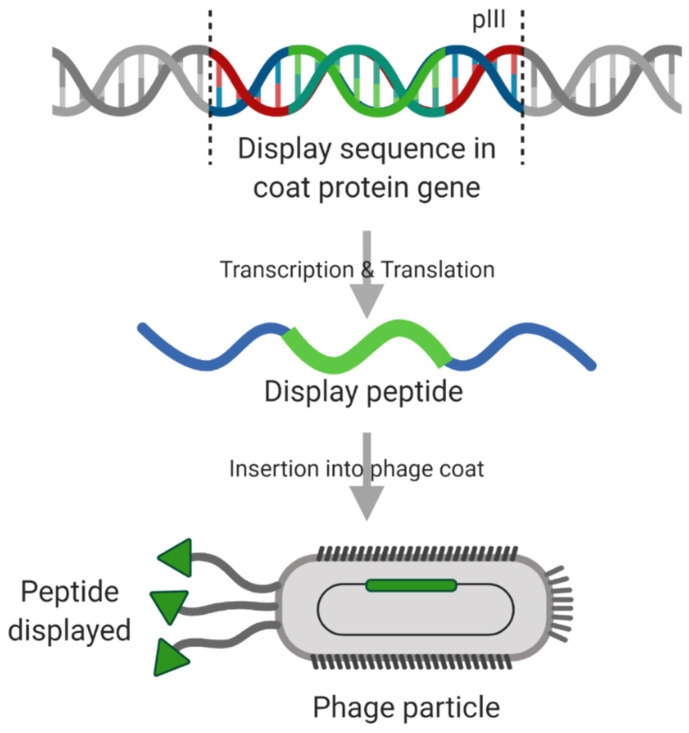
M13-based phage display mechanism. Created with https://biorender.com.

**Figure 8 cancers-12-03327-f008:**
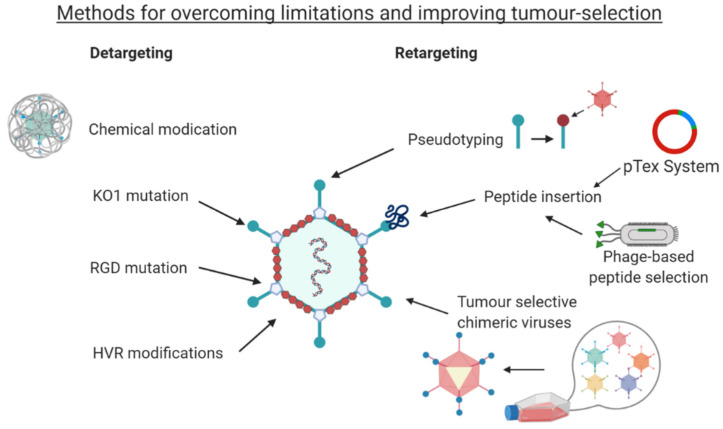
Overview of putative adenoviral detargeting and retargeting approaches. Created with https://biorender.com.

**Table 1 cancers-12-03327-t001:** A summary table of Ads in clinical trials, with their base genome, modifications and their target. Data obtained from https://www.cancer.gov/ (Accessed on: 30 October 2020).

Biologic	Synonyms	Adenovirus genome	Modifications	Targeting	NCI Identifier
GM-CSF-encoding Oncolytic Adenovirus CGTG-102	ONCOS-102	Adenovirus serotype 5/3 (capsid-modified)	Ad5 capsid protein replaced with Ad3 knob domain. Granulocyte-macrophage colony stimulating factor (GM-CSF)	Selective replication in Rb/p16 defective cells. Ad3 receptors.	C98287
OX40L-expressing Oncolytic Adenovirus DNX-2440	Oncolytic Adenovirus Armed with OX40L DNX-2440	Adenovirus serotype 5	Expresses OX40 ligand (OX40L). ∆24 mutation	Selective replication in Rb/p16 defective cells	C160192
Oncolytic Adenovirus ORCA-010	Modified Ad5 ORCA-010	Ad5/3	∆24 mutation. RGD-4C motif. T1 mutation in E3/19K gene	Selective replication in Rb/p16 defective cells. T1 mutation enhances Ad5 release, Ad3 receptors	C168607
	Oncolytic Adenovirus ORCA-010				
Oncolytic Adenovirus Encoding GM-CSF	CG0070	Adenovirus serotype 5	E2F-1 promotor. Granulocyte-macrophage colony stimulating factor (GM-CSF) in E3 region	Selective replication in Rb/p16 defective cells	C48412
Delolimogene Mupadenorepvec	Double-armed TMZ-CD40L/4-1BBL Oncolytic Ad5/35 Adenovirus LOAd703	Adenovirus serotype 5 with L5 segment of fiber replaced with Ad35 fiber	Expresses trimerized CD40 ligand. ∆24 mutation in E1A	Targets CD46. Selective replication in Rb/p16 defective cells	C148462
Oncolytic Adenovirus ICOVIR5-infected Autologous Mesenchymal Stem Cells	LOAd 703	Wildtype human adenovirus 5	RGD-4C motif allows integrin binding. ∆24 in E1A prevents Rb complex and transition into S phase	Bone marrow-derived MSCs target and deliver adenovirus to tumour	C107160
Tasadenoturev	DNX-2401	Adenovirus serotype 5	RGD-4C motif allows integrin binding. ∆24 in E1A prevents Rb complex and transition into S phase	CAR independent. Selective replication in Rb/p16 defective cells	C74067
	(Oncolytic Adenovirus) Ad5-∆24RGD				
	Oncolytic Adenovirus Ad5-DNX-2401				
Tasadenoturev-infected Allogeneic Bone Marrow-derived Mesenchymal Stem Cells	Ad5-DNX-2401-infected Allogeneic Bone Marrow Mesenchymal Stem Cells	Ad5-DNX-2401	RGD-4C motif, ∆24 in E1A prevents Rb complex and transition into S phase	Bone marrow-derived MSCs target and deliver adenovirus to tumour	C159798
	(Allogeneic) BM-hMSC-∆24				
	(Allogeneic) BM-hMSC-∆24-RGD				
Ad5-yCD/mutTKSR39rep-hIL12	Oncolytic Adenovirus Ad5-yCD/mutTKSR39rep-hIL12	Adenovirus serotype 5	Encodes murine interleukin-12 (IL-12) gene in E3 region and a suicide fusion gene (yCD/HSV-1 TKSR39) in E1 region	E1B55K deletion	C123930
Enadenotucirev	ColoAd-1	Chimeric Oncolytic Adenovirus Ad3/Ad11p	Deletions in E3 Region (2444 bp) and E4 Region (24 bp) and 197 Non-homologous nucleotides in the E2B Region	Not fully understood	C113786
	EnAd				

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
