# Peer review of "Hitting the Target but Missing the Point: Recent Progress towards Adenovirus-Based Precision Virotherapies"

_cancers, 2020, doi:10.3390/cancers12113327_

Round 1

Reviewer 1 Report

The manuscript constitutes a very complete and well-documented overview of the field of oncolytic adenoviruses. However it requires some reorganization to be of value for a naive reader.

Major comments

  • My principal concern with the manuscript is its overall lack of structure. For example part 3) Genetic engineering approaches to optimize tumor selectivity describes adenovirus-derived vectors and also principles governing the development of conditionally replicative adenovirus (CRAd) while this notion of CRAd is presented at the beginning of the manuscript (parts 1.3 and part 2 Current applications of oncolytic adenoviral therapies). I suggest to reorganize the review to improve its clarity.
  • It appears that Part 2 Current clinical applications of oncolytic adenoviral therapies reports current clinical applications but also reports many other things such as retargeted and detargeted vectors (lines 217 to 240). I have difficulty to follow the organization of this part.
  • In different parts of the manuscript and in particular parts 5 Retargeting strategies, it is not clear whether the authors are reporting data about replication-deficient vectors or CRAd. This should be clearly stated. For example, see lines 381 to 388 but also other parts of the manuscript.
  • For the paragraph 5.3° Bacteriophage-based technology, I suggest to change the title for something such as Methods to identifiy targeting peptides. Indeed bacteriophage-based technology is just a means by which potential targeting peptides were identified.
  • Regarding the vector Ad5NULL the authors states that it prevents sequestration by and transduction of liver hepatocytes (Uusi-Kertula Clinical Cancer Research 2018). After carefull reading of the reference, I found the authors analyzed the level of DNA and transgene expression 3 days following gene transfer. They found reduced level of DNA and transgene expression compared to a control vector. However, I do not agree to infer a reduced liver uptake from these data. This should be moderated since one could understood that Ad5NULL is available to be retargeted to tumor after intravenous administration.
  • A background about adenovirus receptors, endocytosis, trafficking is needed at the beginning of the review and thus before part 3.

Minor comments

  • All figure legends are too much concise, just a title and small sentence. I think a description of each figure should be included to help the reader to follow the figure.
  • Page 9, lines 301-304, the sentence seems to be incomplete.
  • Reference Hoefherr (line 363) is cited in the text but not found in the bibliography.
  • I do not understand why the authors put the subtitle Intracellular approaches (line 277) in part Genetic engineering approaches to optimize tumor selectivity. This not fits with the content of the paragraph.

Author Response

The manuscript constitutes a very complete and well-documented overview of the field of oncolytic adenoviruses. However it requires some reorganization to be of value for a naive reader.

Author response: We thank the reviewer for their positive scoring of our manuscript, and for their complementary comments regarding the overall quality of the review.

Major comments

  • My principal concern with the manuscript is its overall lack of structure. For example part 3) Genetic engineering approaches to optimize tumor selectivity describes adenovirus-derived vectors and also principles governing the development of conditionally replicative adenovirus (CRAd) while this notion of CRAd is presented at the beginning of the manuscript (parts 1.3 and part 2 Current applications of oncolytic adenoviral therapies). I suggest to reorganize the review to improve its clarity.
  • It appears that Part 2 Current clinical applications of oncolytic adenoviral therapies reports current clinical applications but also reports many other things such as retargeted and detargeted vectors (lines 217 to 240). I have difficulty to follow the organization of this part.

Author response: We have dealt with the above two comments by reorganising the structure of the manuscript, in line with the reviewer comments. We think the revised organisation improves the readability of the review and hope the reviewer agrees.

  • In different parts of the manuscript and in particular parts 5 Retargeting strategies, it is not clear whether the authors are reporting data about replication-deficient vectors or CRAd. This should be clearly stated. For example, see lines 381 to 388 but also other parts of the manuscript.

Author response: since the retargeting strategies described can be applied in both the context of CRAd or replication deficient vectors, we have addressed this point by adding a sentence to the start of section 5 outlining that these approaches can be applied in both contexts and are therefore overviewed together.

  • For the paragraph 5.3° Bacteriophage-based technology, I suggest to change the title for something such as Methods to identifiy targeting peptides. Indeed bacteriophage-based technology is just a means by which potential targeting peptides were identified.

Author response: We agree and have changed the subheading to “Techniques for targeting peptide discovery”

  • Regarding the vector Ad5NULL the authors states that it prevents sequestration by and transduction of liver hepatocytes (Uusi-Kertula Clinical Cancer Research 2018). After carefull reading of the reference, I found the authors analyzed the level of DNA and transgene expression 3 days following gene transfer. They found reduced level of DNA and transgene expression compared to a control vector. However, I do not agree to infer a reduced liver uptake from these data. This should be moderated since one could understood that Ad5NULL is available to be retargeted to tumor after intravenous administration.

Author response: We added an additional reference that evidences that Ad5NULL-A20 can target αvβ6 positive tumours following intravenous administration.

  • A background about adenovirus receptors, endocytosis, trafficking is needed at the beginning of the review and thus before part 3.

Author response: This is an excellent suggestion; we thank the reviewer. Such a section has now been added (1.3.1 – Adenovirus cell entry and trafficking), together with an additional figure.

Minor comments

  • All figure legends are too much concise, just a title and small sentence. I think a description of each figure should be included to help the reader to follow the figure.
  •  

Author response: We expanded the figure legends

  • Page 9, lines 301-304, the sentence seems to be incomplete.

Author response: We amended the sentence to improve it.

  • Reference Hoefherr (line 363) is cited in the text but not found in the bibliography.

Author response: We apologise for this oversight and added the reference.

  • I do not understand why the authors put the subtitle Intracellular approaches (line 277) in part Genetic engineering approaches to optimize tumor selectivity. This not fits with the content of the paragraph.

Author response: We have removed this subtitle.

Please note: our resubmission is provided containing track changes: to view in a more readable format select Review -> No Markup (drop down next to track changes). To view changes, select "All Markup"

Reviewer 2 Report

Well written review on the current status of Adenoviral based virotherapies.

In the Graphical Abstract along side the Phage-display based peptide selection the authors should also include the pFEX sytem described by Lupold et al in the Nucleic Acids Res. 2007;35(20):e138 article.

In the retargeting strategies section, the authors describe some of the limitations of trying to incorporate targets identified and mentioning the need to develop technologies capable of high throughput evolution & selection of knob variants. Lupold et al describes a system that takes advantage of the Adenovirus itself in selecting new variants to selectively bind targets, and can also be combined with reverse bio-panning to select out nonspecific targeting. Nucleic Acids Res. 2007;35(20):e138.

In line 180 there is a minor grammatical error – the authors most probably wanted to say “there is evidence” instead of “there evidence”.

One of the major criticisms is the authors should try to give credit to the researchers that have 1st described a process instead of referencing one of the latest articles, which in turn has referenced another article, and you have to go searching through multiple layers to find the the paper that actually describes what is being tried to be referenced.

e.g. E1A deletion results in replication deficient virus, was 1st described by Frank Graham.   J Gen Virol. 1984 Mar;65 ( Pt 3):585-97. doi: 10.1099/0022-1317-65-3-585.

Also for the 293 cells Dr. Graham should be given the credit for stablishing the cell lines. J Gen Virol. 1977 Jul;36(1):59-74. doi: 10.1099/0022-1317-36-1-59.

For PER.C6 cells were 1st described by Fallaux FJ et al. Hum Gene Ther. 1998 Sep 1;9(13):1909-17. doi: 10.1089/hum.1998.9.13-1909. PMID: 9741429

Description of CRAds, one of the 1st tissue specific oncolytic Adeno virus was described by Rodriguez et al. Cancer Res. 1997 Jul 1;57(13):2559-63., and the concept was described by Berns KI in 1995. Ann N Y Acad Sci. 1995 Nov 27;772:95-104.

Author Response

Well written review on the current status of Adenoviral based virotherapies.

Author response: We thank the reviewer for the good scores of our review and positive comments.

In the Graphical Abstract along side the Phage-display based peptide selection the authors should also include the pFEX sytem described by Lupold et al in the Nucleic Acids Res. 2007;35(20):e138 article.

In the retargeting strategies section, the authors describe some of the limitations of trying to incorporate targets identified and mentioning the need to develop technologies capable of high throughput evolution & selection of knob variants. Lupold et al describes a system that takes advantage of the Adenovirus itself in selecting new variants to selectively bind targets, and can also be combined with reverse bio-panning to select out nonspecific targeting. Nucleic Acids Res. 2007;35(20):e138.

Author response: We thank the reviewer for highlighting this paper. We have now referenced this study, described its findings in detail and amended the graphical abstract to include this technology.

In line 180 there is a minor grammatical error – the authors most probably wanted to say “there is evidence” instead of “there evidence”.

Author response: We have corrected this. 

One of the major criticisms is the authors should try to give credit to the researchers that have 1st described a process instead of referencing one of the latest articles, which in turn has referenced another article, and you have to go searching through multiple layers to find the the paper that actually describes what is being tried to be referenced.

e.g. E1A deletion results in replication deficient virus, was 1st described by Frank Graham.   J Gen Virol. 1984 Mar;65 ( Pt 3):585-97. doi: 10.1099/0022-1317-65-3-585.

Also for the 293 cells Dr. Graham should be given the credit for stablishing the cell lines. J Gen Virol. 1977 Jul;36(1):59-74. doi: 10.1099/0022-1317-36-1-59.

For PER.C6 cells were 1st described by Fallaux FJ et al. Hum Gene Ther. 1998 Sep 1;9(13):1909-17. doi: 10.1089/hum.1998.9.13-1909. PMID: 9741429

Description of CRAds, one of the 1st tissue specific oncolytic Adeno virus was described by Rodriguez et al. Cancer Res. 1997 Jul 1;57(13):2559-63., and the concept was described by Berns KI in 1995. Ann N Y Acad Sci. 1995 Nov 27;772:95-104.

Author response: We thank the reviewer for highlighting this. We apologise for not citing the appropriate first source, and absolutely agree that this the correct etiquette. We have updated and included the appropriate sources, as highlighted by the reviewer.

Please note: our resubmission is provided containing track changes: to view in a more readable format select Review -> No Markup (drop down next to track changes). To view changes, select "All Markup"

Reviewer 3 Report

Parker and colleagues provides a very nice overview of (adeno)virus targeting strategies in the context of the present research and clinical environment of the virotherapy field. The review is of high quality, particularly interesting - providing a unique perspective of targeting approaches – and up-to-date. Thus, it will certainly be a work of reference for future reports.

I have only minor comments which the authors might find helpful to further improve the manuscript.

  • Title: I was curious about the title, which to me indicated that targeting adenoviruses is technically working (“Hitting the target”), but – surprisingly – that it is of no sense (“but missing the point”). There is no reference what is actually meant in the review. Did I miss/misunderstand something?
  • One topic of controversial debate in the virotherapy community has been, whether systemic application of oncolytic viruses is necessary to achieve systemic therapeutic activity, i.e. for treatment of metastatic disease. In recent years, a high number of excellent pre-clinical and clinical studies have shown that viral oncolysis triggers systemic anti-tumor immunity that attacks non-injected lesions. Therefore, I think the authors’ statement in line 132, that treatment of metastatic disease requires i.v. virus application is too strong.
  • In its present version the table is very difficult to read (small text and low resolution)
  • The paragraph headers are a bit confusing: The Genetic engineering paragraph has intracellular approaches as sole sub-heading, but then entry targeting approaches – also by genetic engineering – are separate paragraphs
  • Line 31: the authors probably mean proteins (not transgenes) encoded by…..
  • Line 143: high capacity adenoviral vectors can incorporate approx.. 36 kb of heterologous DNA
  • Line 263: to me it is not clear what the authors mean with “…..depletion of virotherapy available for active targeting of tumors.”
  • Line 303: this sentence reads like cancer cells have mutated to express viral genes that are deleted in oncolytic viruses for targeting purposes. Re-phrase?
  • De-targeting: fiber chimerism can also be exploited for de-targeting (using Ad40 or Ad41 short fibers)
  • Line 373: do the authors mean chimeric fiber proteins with knob and shaft/tail domain originating from different serotypes (but knobs themselves not chimeric)?
  • Line 384: For clarity, the authors could refer here also to the tail domain’s role in pseudotyping
  • Biopanning: Has also been performed with adenovirus libraries containing randomized peptides in the virus capsid

Author Response

Parker and colleagues provides a very nice overview of (adeno)virus targeting strategies in the context of the present research and clinical environment of the virotherapy field. The review is of high quality, particularly interesting - providing a unique perspective of targeting approaches – and up-to-date. Thus, it will certainly be a work of reference for future reports.

Author response: We thank the reviewer for their excellent scores and kind comment on the quality of the review. We are grateful that the reviewer indicates that they expect the review to become well cited. We agree (but are admittedly biased in this regard!).

I have only minor comments which the authors might find helpful to further improve the manuscript.

  • Title: I was curious about the title, which to me indicated that targeting adenoviruses is technically working (“Hitting the target”), but – surprisingly – that it is of no sense (“but missing the point”). There is no reference what is actually meant in the review. Did I miss/misunderstand something?

Author response: This was a source of some discussion amongst the authors prior to submission and following this comment from the reviewer. We prefer to retain the title. A key aspect to drawing in a reader to a review is a compelling title, which we feel this is. Our point is that the technologies exist for developing targeting agents to tumours (“hitting the target”) but that they currently don’t work brilliantly (possibly with the occasional exception) in the context of the virotherapy (“missing the point”) and that we need to develop smarter tools to target viruses effectively. Our penultimate sentence attempts now to make this point more clearly, bringing the engaged reader back to the title (line 653-657). We did ponder changing this to “Close but no cigar” but we prefer “Hitting the target but missing the point”. If the reviewer feels strongly about this, we can change it, but prefer to retain the title “as is” if possible.

  • One topic of controversial debate in the virotherapy community has been, whether systemic application of oncolytic viruses is necessary to achieve systemic therapeutic activity, i.e. for treatment of metastatic disease. In recent years, a high number of excellent pre-clinical and clinical studies have shown that viral oncolysis triggers systemic anti-tumor immunity that attacks non-injected lesions. Therefore, I think the authors’ statement in line 132, that treatment of metastatic disease requires i.v. virus application is too strong.

Author response: the reviewer makes a good point, and we have added words to this effect, as well as references to describe such abscopal effects.

  • In its present version the table is very difficult to read (small text and low resolution)

Author response:  Unfortunately, although the table is rather crowded for one page, it is too small to be spread over two pages. We have therefore improved the resolution of the table in the hope that it makes the table more readable.

  • The paragraph headers are a bit confusing: The Genetic engineering paragraph has intracellular approaches as sole sub-heading, but then entry targeting approaches – also by genetic engineering – are separate paragraphs

Author response: This is covered in our response to reviewer one. The subheadings have now been changed.

  • Line 31: the authors probably mean proteins (not transgenes) encoded by…..

Author response: This has been corrected.

  • Line 143: high capacity adenoviral vectors can incorporate approx.. 36 kb of heterologous DNA

Author response: We added a sentence (see line 255-8) to mention gutless/HDAds, and a reference, but as their use it highly limited for oncology applications, we have not expanded further.

  • Line 263: to me it is not clear what the authors mean with “…..depletion of virotherapy available for active targeting of tumors.”

Author response: We have expanded this to emphasise that we mean that “off-target” uptake via ubiquitously expressed receptors will result in loss of virions to uptake in non-target (i.e. non-tumour) organs.

  • Line 303: this sentence reads like cancer cells have mutated to express viral genes that are deleted in oncolytic viruses for targeting purposes. Re-phrase?

Author response: We rephrased to make clearer

  • De-targeting: fiber chimerism can also be exploited for de-targeting (using Ad40 or Ad41 short fibers)

Author response: We added mention of this (lines 379-381) and references (125-127)

  • Line 373: do the authors mean chimeric fiber proteins with knob and shaft/tail domain originating from different serotypes (but knobs themselves not chimeric)?

Author response: We have adjusted the wording in the first paragraph of 5.1 to clarify this point.

  • Line 384: For clarity, the authors could refer here also to the tail domain’s role in pseudotyping

Author response: This is a good point we omitted previously and we thank the reviewer for highlighting this. We include a new sentence to cover this in paragraph 3 of 5.2 (line 408-410)

  • Biopanning: Has also been performed with adenovirus libraries containing randomized peptides in the virus capsid

Author response: We added additional references in this regard (ref 163).

Please note: our resubmission is provided containing track changes: to view in a more readable format select Review -> No Markup (drop down next to track changes). To view changes, select "All Markup"

Round 2

Reviewer 1 Report

The authors took in account all my remarks and improved the organization of their manuscript. I think the manuscript is now suitable for publication in its present form in Cancers. Congratulations to the authors for their work.